# Treatments of COVID-19-Associated Taste and Saliva Secretory Disorders

**DOI:** 10.3390/dj11060140

**Published:** 2023-05-25

**Authors:** Hironori Tsuchiya

**Affiliations:** Department of Dental Basic Education, Asahi University School of Dentistry, Mizuho, Gifu 501-0296, Japan; tsuchi-hiroki16@dent.asahi-u.ac.jp; Tel.: +81-58-329-1263

**Keywords:** COVID-19, taste disorder, saliva secretory disorder, treatment, pathogenic mechanism

## Abstract

Since the worldwide spread of severe acute respiratory syndrome coronavirus 2 (SARS-CoV-2) infection, treating taste and saliva secretory disorders associated with coronavirus disease 2019 (COVID-19) has become a critical issue. The aim of the present study was to update information on treatments applicable to such oral symptoms and discuss their pathogenic mechanisms. The literature search indicated that different treatments using tetracycline, corticosteroids, zinc, stellate ganglion block, phytochemical curcumin, traditional herbal medicine, nutraceutical vitamin D, photobiomodulation, antiviral drugs, malic acid sialagogue, chewing gum, acupuncture, and/or moxibustion have potential effects on COVID-19-associated ageusia/dysgeusia/hypogeusia and xerostomia/dry mouth/hyposalivation. These treatments have multiple modes of action on viral cellular entry and replication, cell proliferation and differentiation, immunity, and/or SARS-CoV-2 infection-induced pathological conditions such as inflammation, cytokine storm, pyroptosis, neuropathy, zinc dyshomeostasis, and dysautonomia. An understanding of currently available treatment options is required for dental professionals because they may treat patients who were infected with SARS-CoV-2 or who recovered from COVID-19, and become aware of their abnormal taste and salivary secretion. By doing so, dentists and dental hygienists could play a crucial role in managing COVID-19 oral symptoms and contribute to improving the oral health-related quality of life of the relevant patients.

## 1. Introduction

Severe acute respiratory syndrome coronavirus 2 (SARS-CoV-2), belonging to the family *Coronaviridae*, is an enveloped virus with a positive-sense, single-stranded RNA genome that encodes major structural proteins involving envelope, membrane, nucleocapsid, and spike protein. Since this novel RNA virus was identified as a causative agent of coronavirus disease 2019 (COVID-19), infection with SARS-CoV-2 has quickly spread worldwide, with the continuous appearance of many different variants. As of 10 March 2023, over 676.6 million individuals were globally infected with SARS-CoV-2 and over 6.8 million patients with COVID-19 had died in the world, according to the Johns Hopkins University and Medicine Coronavirus Resource Center [1]. Worldometer, a reference website of live world statistics [2], indicates over 688.3 million COVID-19 cases and over 660.7 million recovered cases as of 15 May 2023. Over three years have passed since the emergence of COVID-19 in Wuhan, China, and it has become apparent that SARS-CoV-2 infection produces a wide range of manifestations, varying from asymptomatic presentation to respiratory distress syndrome, and death in a critical case. Fever, cough, dyspnea, myalgia, and cardiomyopathy are among the most prevalent symptoms in the early stage of SARS-CoV-2 infection. In addition to them, COVID-19 is characterized by oral symptoms: ageusia (taste loss), dysgeusia (taste impairment), and hypogeusia (taste reduction) [3], and xerostomia, dry mouth (subjective complaint of oral dryness), and hyposalivation (objective reduction in salivary flow) [4]. Up to 93% of COVID-19 patients present with taste disorders, with a prevalence of 3–67% for ageusia, 19–48% for dysgeusia, and 35% for hypogeusia [5,6,7,8,9]. Saliva secretory disorders have also been reported by 42–77% of COVID-19 patients [6,7,8,10,11]. Even after complete vaccination (Pfizer/BioNTech, AstraZeneca, Moderna, and Johnson & Johnson vaccines), 54% of patients who developed symptomatic COVID-19 complained of oral symptoms [12].

COVID-19 oral symptoms were initially considered to be transient and were assumed to eventually disappear without treatment. However, there is increasing evidence that taste and saliva secretory disorders persist in subjects who recovered from COVID-19, as well as other sequelae such as fatigue, dyspnea, cough, headache, neurocognitive impairment (or brain fog), and hair loss. Persistent taste disorders have been reported by many COVID-19 survivors who were followed up after recovery from the disease [13,14]. The prevalence was 39% for dysgeusia at an 8-month follow-up [11], 19% for ageusia at a 12-month follow-up [15], and 22% for taste or smell impairment at a 12-month follow-up [16]. According to follow-up studies [11,17,18] and comprehensive symptom reviews [13,19], xerostomia, dry mouth, and hyposalivation persist in COVID-19 survivors, with a prevalence of 9–29% at 1–8 month follow-ups. Parageusia (wrong taste elicited by a taste stimulus) and phantogeusia (feeling taste in the absence of a taste stimulus) have also been observed in 23.5% and 17.6%, respectively, of subjects followed up 2 months after a negative reverse transcription-polymerase chain reaction (RT-PCR) test [20].

Although taste and saliva secretory disorders are not life-threatening, their persistence not only reduces the oral-health-related quality of life of COVID-19 patients and survivors [21], but adversely affects their nutrition by decreasing appetite [22], and even elevates the risk of depression and suicidal ideation [23]. A reduction in salivary secretion is linked to the increased incidence of dental caries, periodontal disease, oral infection, and halitosis, and to difficulty experienced in mastication, swallowing, and speaking [24]. Therefore, treating COVID-19-associated taste and saliva secretory disorders has become one of the most critical issues in the COVID-19 era. The aim of the present study was to update information on treatments applicable to such oral symptoms in the early stage of SARS-CoV-2 infection and after recovery from the disease, and to discuss the pathogenic mechanisms underlying them.

Neta et al. [25] and Khani et al. [26] recently reviewed pharmacological approaches to potential treatments of smell and taste disorders in COVID-19 patients. Although gustation is affected when there is an abnormality in olfaction, taste disorders are more prevalent than smell disorders, as reported by several COVID-19 symptomatology studies [27,28,29,30], and COVID-19 ageusia is not necessarily accompanied by anosmia with nasal obstruction and rhinitis [31]. Taste disorders are observed in almost all COVID-19 patients with smell disorders, whereas only 30% of abnormal taste cases show smell disorders [32]. A taste disorder could be referred to as an independent symptom rather than one of the COVID-19 symptoms closely associated with olfactory dysfunction [33]. Taste disorders frequently co-occur with saliva secretary disorders in COVID-19 patients and survivors [11,13], while taste disorders are more prevalent [7,34] or less prevalent than saliva secretory disorders [6,35]. The present study focused on oral symptoms and their treatments applied to human subjects. Neither animal experiments nor in silico studies were covered.

## 2. Materials and Methods

In order to collect information on treatments applicable to COVID-19 oral symptoms, a literature search was performed in PubMed, LitCovid, ProQuest, and Google Scholar with a cutoff date of 31 March 2023. Given the constant progression of studies on COVID-19, the preprint databases medRxiv and bioRxiv were also used to refer to the most up-to-date data. The literature search was conducted by using the following terms and combinations thereof: “COVID-19 treatment”, “ageusia”, “dysgeusia”, “hypogeusia”, and “COVID-19 therapy” for treatments of COVID-19-associated taste disorders; and “COVID-19 treatment”, “xerostomia”, “dry mouth”, “hyposalivation”, and “COVID-19 therapy” for treatments of COVID-19-associated saliva secretory disorders. The exclusion criteria were papers that were not published in English, had not been peer reviewed, and were not from academic sources. Cited papers in the retrieved articles were further searched for additional references.

## 3. Results and Discussion

Specific treatments for taste and saliva secretory disorders can be designed and developed through understanding their pathophysiology. However, only hypotheses have been proposed so far to explain how SARS-CoV-2 infection causes ageusia, dysgeusia, or hypogeusia and xerostomia, dry mouth, or hyposalivation [3,4,10,27,36,37]. While it is very likely that COVID-19 oral symptoms are the result of local or systemic etiology, or both, their pathogenic mechanisms remain largely unclear. The treatment strategy consists of (1) targeting SARS-CoV-2 that can enter, replicate in, and cause damage to cells responsible for taste perception and salivary secretion; (2) alleviating, reducing, or protecting against SARS-CoV-2 infection-induced pathological conditions such as inflammation, cytokine storm, pyroptosis, neuropathy, zinc dyshomeostasis, and dysautonomia; (3) symptomatic therapies for taste and saliva secretory disorders; and (4) alternative medicines to be expected from the clinical outcomes of applied studies.

In the literature, there have not been many clinical trials conducted with COVID-19 patients and survivors as the subjects, especially for saliva secretory disorder treatments. Therefore, in addition to outcomes of interventional and observational studies in COVID-19 cases, clinical case reports and promising effects on non-COVID-19 patients or subjects were used for discussion in the following sections.

### 3.1. Treatments of COVID-19-Associated Taste Disorders

Diverse treatment approaches have been tried for ageusia, dysgeusia, or hypogeusia caused by SARS-CoV-2 infection. In the following Sections, different types of treatments applicable to COVID-19-associated taste disorders are reviewed together with discussions on their pathogenic mechanisms and treatment rationales. They are summarized in Table 1, with the methods used, patients or subjects, clinical outcomes, and references included.

#### 3.1.1. Tetracycline

Viral Cellular Entry, Inflammatory Cell Death, and Neuropathy

For SARS-CoV-2 to enter host cells, its spike protein binds to a cellular receptor angiotensin-converting enzyme 2 (ACE2) through the receptor binding domain (RBD), followed by viral and cellular membrane fusion that is mediated by cellular protein convertase (Furin) and transmembrane serine protease 2 (TMPRSS2) [54]. While ACE2 is ubiquitously distributed in various tissues and organs, such as the lung, heart, kidney, liver, digestive tract, brain, and nervous system, it is also richly expressed in the taste buds and taste bud-embedded papillae of humans [55]. Each taste bud consists of 50–100 packed specialized epithelial cells, taste receptor cells, that include distinct types of cells such as type-I cells (glia-like supporting cells) to respond to salty stimuli, type-II cells (G-protein coupled receptors) to respond to sweet, bitter, and umami stimuli, and type-III cells (ion channels) to respond to sour stimuli. Lingual papillae are classified into circumvallate, fungiform, and foliate papillae, all of which contain taste buds, and filiform papillae being devoid of taste buds but involved in the texture perception of foods. Doyle et al. [56] revealed that ACE2 receptors are expressed in taste receptor type-II cells within taste buds buried in circumvallate and fungiform papillae, and that replicating SARS-CoV-2 is present in type-II cells of fungiform papillae biopsied from COVID-19 patients with taste disorders. The specific expression of ACE2 and the presence of SARS-CoV-2 are consistent with the characteristics of taste disorders, i.e., the sweet, bitter, and umami tastes are more frequently impaired in moderate COVID-19 [57], but the salty taste is less significantly affected by COVID-19 [58]. Since SARS-CoV-2 interacts with ACE2 receptors expressed in taste buds and papillae, it is conceivable that cytopathic SARS-CoV-2 damages these tissues responsible for taste perception, causing their dysfunctions in the process of viral cellular entry [59,60]. Since viral shedding is found for extended periods after the recovery from COVID-19, the long-lasting presence of SARS-CoV-2 in taste buds could cause the persistence of taste disorders in COVID-19 survivors.

Following the cellular entry of SARS-CoV-2, the relevant cells undergo a highly inflammatory cell death, pyroptosis, that triggers the generation and secretion of pro-inflammatory cytokines and chemokines, promoting further inflammation [61]. SARS-CoV-2 infection induces the inflammation of taste receptor cells and taste buds to impair taste perception. Their inflammatory responses could cause damage to cells responsible for multiple tastes (sour, salty, sweet, bitter, and umami taste) [57]. While mammalian caspases are classified into apoptotic and inflammatory caspases, caspase-1, belonging to the latter, is activated to induce pyroptotic cell death and release pro-inflammatory cytokines [62]. Inflammatory cytokines also trigger apoptotic cell death, resulting in the abnormal turnover of taste buds.

Since neurotropic SARS-CoV-2 has neuro-invasive potential, viral infection-induced neuropathy is pathogenically related to COVID-19-associated taste disorders [63]. Taste signals are transmitted to the brain stem through taste bud-innervating cranial nerves. The facial nerve (cranial nerve VII), the glossopharyngeal nerve (cranial nerve IX), and the vagus nerve (cranial nerve X) innervate the anterior two-thirds of the tongue, the posterior one-third of the tongue, and the epiglottis region, respectively, to transmit information on tastants. Among them, the facial nerve is most commonly affected by COVID-19, followed by the glossopharyngeal nerve and the vagus nerve [64]. A possible link has been indicated between COVID-19 and facial nerve paralysis [65]. Taste disorders are also accompanied by neurodegeneration occurring in COVID-19 cases [66], and COVID-19 promotes neurodegenerative changes [67]. While a significant proportion of COVID-19 patients present with persistent taste disorders even after two consecutive negative nasopharyngeal swabs [68], such long-term sequelae could be related to neural damages due to SARS-CoV-2 infection.

Drug repurposing is a strategy which uses existing and approved drugs for novel therapeutic purposes, and which has the advantage of reducing costs and the time needed for developing new drugs and using drugs with confirmed safety and commercial availability. Among the repurposed drugs for COVID-19 are tetracyclines, which exhibit viral cellular entry-suppressing, viral replication-inhibiting, anti-inflammatory, neuroprotective, and anti-apoptotic activities in addition to the antibiotic activity. Tetracyclines have antiviral effects on SARS-CoV-2 that could enter taste cells, replicate in them, and affect their functions [69]. Tetracyclines have so high an affinity for the RBD of SARS-CoV-2 that they inhibit the binding between the viral spike protein and the ACE2 receptor [70]. In in vitro experiments using Vero E6 cells infected with SARS-CoV-2, doxycycline inhibited viral entry and replication at low micromolar concentrations [71]. Doxycycline also possesses the anti-inflammatory property to inhibit pro-inflammatory cytokines such as interleukin (IL)-6 and tumor necrosis factor (TNF)-α [72]. Minocycline has been reported to inhibit the expression and activation of caspase-1 expressed by SARS-CoV-2 infection and counteract cytokine storm inflammation in COVID-19 [73]. Tetracyclines, especially the second-generation minocycline and doxycycline, are expected to be effective in the treatment of taste disorders associated with inflammatory responses to SARS-CoV-2 infection. In addition to antiviral and anti-inflammatory effects, tetracyclines exhibit an anti-apoptotic activity that is protective against neurological disorders [74]. Minocycline and doxycycline also have multimodal neuroprotective effects [75].

2.Treatment with Tetracyclines and Outcomes

Gironi et al. [38] conducted a multicenter prospective observational study to evaluate the effects of tetracyclines on COVID-19 symptoms. They orally administered either doxycycline (100 mg once a day (n = 10) or 100 mg twice a day (n = 15)) or minocycline (50 mg once a day (n = 8), 100 mg once a day (n = 3), or 100 mg twice a day (n = 2)) to COVID-19 outpatients (n = 38, female: 52.6%, age: 21–67 years). Ageusia reported by 23.7% of the patients disappeared in the first week of treatment. Other symptoms, including fatigue and dyspnea, were also resolved in all patients within 10 days.

#### 3.1.2. Corticosteroid

Inflammation of the Taste Buds and Papillae

SARS-CoV-2 infection induces pro-inflammatory cytokines such as IL-6, IL-8, and TNF-α. Among them, IL-6 acts on taste receptor cells to cause inflammation, and there is a close relationship between IL-6 increase and taste disorder occurrence [76]. Multiple tastes are impaired to cause ageusia in COVID-19 patients with high IL-6 levels [57]. While inflammation activates the interferon (INF) signaling pathway in taste bud cells, viral infection induces INFs that not only impair the function of taste buds to disturb taste transduction, but cause the apoptotic cell death of taste buds to affect the turnover and renewal of different types of taste cells [77]. Henin et al. [78] performed morphological and histopathological examinations of tongues collected from cadavers. Their RT-PCR analyses indicated that SARS-CoV-2 was present in circumvallate and foliate papillae, both of which showed severe inflammation (infiltrate and fibrosis) with the destruction of taste buds. They speculated that ACE2 and INF-γ mediate inflammatory damages to these papillae. Circumvallate and foliate papillae could be invaded by SARS-CoV-2 to produce inflammation, with the subsequent damage to taste buds resulting in taste disorders.

Since SARS-CoV-2 infection is related to inflamed taste buds and papillae, an anti-inflammatory corticosteroid is usable for treating COVID-19-associated taste disorders.

2.Treatment with Corticosteroid and Outcomes

In a prospective interventional study by Singh et al. [39], COVID-19 patients (n = 60, female: 25.0%, mean age: 50.9 years) complaining of dysgeusia and anosmia due to mild to moderate disease were subjected to the local application of triamcinolone oral paste, which commonly contains 0.1% triamcinolone acetonide. The taste sensation of the patients was tested by applying different tastant solutions to the anterior two-thirds of the tongue. Control subjects (n = 60, female: 28.3%, mean age: 51.2 years) did not receive any interventions. All study participants in both groups had dysgeusia. Sweet, bitter, salty, and sour tastes were improved in 91.7%, 83.3%, 83.3%, and 86.6% of patients, respectively, on day 5 of treatment, whereas taste disorders of control subjects further worsened on day 5 compared with day 1. Triamcinolone used in this study acts only on the site of application, with minimal systemic side-effects [79]. The use of triamcinolone oral paste could be one of treatment options for COVID-19-associated taste disorders [80].

Gamil et al. [40] conducted a prospective cohort study to confirm the effect of systemic corticosteroids on COVID-19-associated taste disorders. Among a total of 80 study participants (female: 51.3%, age: 18–67 years), 67 COVID-19 patients who were diagnosed with the RT-PCR test and who presented with ageusia were recruited. All patients were prescribed 10 mg corticosteroid once a day in the early morning for the first week and then oral corticosteroid was reduced to 5 mg in the second week to withdraw the drug gradually. The patients were followed up weekly for 3 months. According to the duration of taste disorders, patients were assigned into two groups: the early group (within 1 week; n = 34, female: 55.9%) and the late group (>1 week; n = 33, female: 57.6%). All patients recovered from ageusia at the end of the treatment and reported no side-effects of the corticosteroid. While the recovery time was different between the early and late groups, the systemic administration of corticosteroid was effective for the treatment of COVID-19 ageusia as well as in a local application.

From a perspective of inflammation suppression, non-steroidal anti-inflammatory drugs may be applicable to COVID-19-associated taste disorders. However, their use has not been recommended because of the possibility of aggravating COVID-19 symptoms [81]. In a case report, an adult male COVID-19 patient with reactive arthritis and ageusia orally took ibuprofen of 400 mg twice a day [82]. Consequently, his arthritis-relating symptom was resolved 2 days after treatment, but ageusia remained.

#### 3.1.3. Zinc

Zinc Deficiency Induced by SARS-CoV-2 Infection

Zinc plays an important role, not only in physiological functions at a level of taste buds and taste stimulus-transmitting nerves, but also in the regeneration and maintenance of taste cells. Since maintaining the appropriate concentration of intracellular zinc is essential for the functional and morphological normality of cells, zinc deficiency has an adverse impact on papillae, taste buds, and taste receptor cells to decrease their number and size [83]. COVID-19 patients show hypozincemia [13,84], in which zinc is redistributed from blood to the liver at the expense of zinc in peripheral tissues and cells, as observed in infection-induced systemic inflammation [85]. SARS-CoV-2 infection disturbs zinc homeostasis, or causes zinc dyshomeostasis, in cells responsible for taste perception [86]. Taste disorders are pathogenically related to zinc deficiency resulting from SARS-CoV-2 infection [13]. Among the different isozymes of zinc-metalloenzyme carbonic anhydrases, carbonic anhydrase VI is localized in the taste buds and secreted into the saliva from the parotid and submandibular glands [87]. Carbonic anhydrase VI is a trophic factor to promote the growth, development, and maintenance of taste buds and papillae. Henkin et al. [88] suggested that a decrease of carbonic anhydrase VI in parotid saliva is associated with a pathological change of the taste buds. COVID-19 patients with ageusia or hypogeusia showed significantly low levels of salivary zinc, which were elevated with recovery from taste disorders [89]. When patients suffering from taste disorders were supplemented with 100 mg zinc once a day for 4–6 months, they showed significant increases in salivary zinc and carbonic anhydrase VI, together with the morphological recovery of taste buds and the resolution of taste disorders [90]. Zinc-binding metallothionein-3 is also expressed in taste buds [91]. The formation and development of taste buds depends on afferent nerves and sensory neurite growth [92]. If SARS-CoV-2 infection upregulates the expression of metallothionein-3, cellular metallothionein-3 would increase and affect taste buds because it has the ability to inhibit neurite formation.

Given the SARS-CoV-2 infection-induced disturbance of zinc homeostasis causing zinc deficiency in taste buds and papillae, zinc supplementation to compensate for deficient zinc is expected to be one of treatments for COVID-19-associated taste disorders [13,93]. Zinc is also able to inhibit viral replication, which should contribute to increasing the efficacy of zinc supplementation.

2.Supplementation with Zinc and Outcomes

Abdelmaksoud et al. [41] assessed the effects of zinc on chemosensory disorders associated with COVID-19. COVID-19 patients (n = 49) with different disease severity orally received 220 mg zinc sulfate (corresponding to elemental zinc of 50 mg) twice a day and they were followed up until pharyngeal swabs became SARS-CoV-2 negative. The patients showed a shorter duration of recovery of tase and smell functions than control subjects (n = 56) without zinc supplementation, although further details were not mentioned about taste improvement.

In a case report by Finzi [42], COVID-19 outpatients (n = 4) were orally supplemented with zinc citrate (corresponding to elemental zinc of 23 mg), zinc citrate/zinc gluconate (23 mg), or zinc acetate (15 mg), which were taken as lozenges every 2–4 h by dissolving on the tongue over 20–30 min. Such zinc supplementations showed symptomatic and objective improvements in all patients. However, this report did not refer to detailed effects on oral symptoms.

Ben Abdallah et al. [43] performed a randomized double-blind controlled trial, in which ambulatory and hospitalized COVID-19 patients (n = 231, female: 47.6%, mean age: 54.6 years) orally received 25 mg of elemental zinc twice a day for 15 days. They revealed that zinc supplementation shortened the symptom duration, but did not mention in detail its effect on taste disorders.

Finzi and Harrington [44] performed a retrospective study in which 21 and 7 COVID-19 patients (a total of n = 28, female: 60.7%, mean age: 40 years) orally received zinc gluconate/citrate-containing lozenges (corresponding to elemental zinc of 23 mg) and zinc acetate-containing lozenges (corresponding to elemental zinc of 15 mg), respectively, at a total dosage of 2–2.5 mg/kg/day by taking 6–12 lozenges once a day depending on weight. COVID-19 symptoms, including taste and smell loss, were assessed by scoring. The symptom scores were significantly decreased 7 days after zinc supplementation, and the symptomatic improvement began mean 1.6 days after treatment. Zinc gluconate was found to be better tolerated than zinc acetate.

Citu et al. [45] assessed the effects of mineral supplementation during pregnancy. In a cohort of 448 pregnant patients with COVID-19, 74 patients were supplemented with a combination of zinc, magnesium, and calcium. Consequently, ageusia and anosmia were reported by 41.9% of subjects with zinc supplementation, but by 57.2% of subjects without zinc supplementation.

On the other hand, one randomized clinical trial suggested that supplementation with high-dose zinc gluconate, ascorbic acid, and their combination was not effective in decreasing the duration of symptoms of COVID-19 ambulatory patients compared with the standard care [94].

#### 3.1.4. Stellate Ganglion Block

Dysautonomia

Patients who have recovered from COVID-19 frequently complain of long-lasting neurological symptoms such as fatigue, headache, cognitive impairment, and taste/smell disorders. Many of these symptoms are observed in patients with dysautonomia, the overactivity of sympathetic or parasympathetic components of the autonomic nervous system. Familial dysautonomia is characterized by a high incidence of perception failures of sweet, bitter, and salty tastes [95]. Since the excessive activity of the sympathetic nervous system is implicated in comorbidities of COVID-19 [96,97], it is conceivable that dysautonomia pathophysiologically underlies taste disorders [98].

Stellate ganglion block, which is performed by injecting local anesthetics (lidocaine, bupivacaine, etc.) into or around the stellate ganglion, effectively blocks the activity of the cervical sympathetic chain innervating post-ganglionic neurons. It has been used for the purpose of treating sympathetically mediated pathological conditions, and its safety has been well established. Given the possibility that COVID-19-associated taste disorders are due to the dysregulation of the sympathetic nervous system, stellate ganglion block is expected to be one of treatment options for them.

2.Stellate Ganglion Block and Outcomes

In a case series by Liu and Duricka [46], a 42-year-old female who recovered from COVID-19 but had continued to suffer from taste and smell disorders underwent right-sided stellate ganglion block with a local anesthetic (the drug used was not described). Ipsilateral dysgeusia and anosmia were improved immediately after treatment, but the disorders persisted contralaterally. Therefore, she underwent left-sided stellate ganglion block 2 days later. Taste and smell functions were restored to a normal state at 2-week follow-up. Another 44-year-old female with taste and smell loss, who had contracted SARS-CoV-2 approximately 8 months before, underwent right-sided stellate ganglion block and left-sided stellate ganglion block on the next day. Within minutes of each treatment, she reported the drastic improvement in ipsilateral dysgeusia and anosmia. Restored taste and smell functions were confirmed at 60-day follow-up.

Chauhan et al. [47] carried out stellate ganglion block with 4 mL of 0.25% bupivacaine for a 48-year-old female who had recovered from COVID-19 4 months before but who complained of an altered taste in various types of foods. She underwent right-sided stellate ganglion block and then left-sided stellate ganglion block 3 days later. Consequently, her taste disorders were improved a few days after treatment.

#### 3.1.5. Phytochemical

Multiple Pathogenic Mechanisms

As described in Section 3.1.1 and Section 3.1.2, COVID-19-associated taste disorders are pathogenically interpreted by multiple mechanisms: viral cellular entry by interacting with ACE2 and TMPRSS2 expressed in taste buds and papillae, damages to taste buds and papillae in the process of viral cellular invasion, viral infection-induced inflammation of taste buds and papillae, and viral neuropathy of taste bud-innervating cranial nerves. Persistent inflammation is also related to the weakened antioxidant defense or the imbalanced antioxidant system of COVID-19 patients.

Phytochemicals, bioactive plant components, have antiviral [99], anti-inflammatory [100], neuroprotective [101], and antioxidant activities [102] which are implicated in managing COVID-19 neurological symptoms. SARS-CoV-2 main protease (M^pro^) is essential for the cleavage of viral nonstructural polypeptides into individual functional proteins; therefore, M^pro^ plays an important role in viral replication. A molecular docking study has suggested that phytochemicals such as curcuminoid (curcumin), flavonoid (rutin, hesperidin, (–)-epigallocatechin gallate, (–)-epigallocatechin, quercitrin, etc.), and capsaicinoid (capsaicin) inhibit SARS-CoV-2 M^pro^ [103].

Since phytochemicals have diverse anti-COVID-19 potential, they are usable for treating COVID-19-associated taste disorders. Koyama et al. [104] recently reported a very excellent review on the possible use of phytochemicals for recovery from anosmia and ageusia induced by COVID-19.

2.Treatment with Curcumin and Outcomes

Curcumin derived from the rhizome of turmeric (*Curcuma longa*) has a high binding affinity for SARS-CoV-2 spike protein, host cell ACE2 receptor, and RBD/ACE2 complex [105], suggesting that curcumin may inhibit the cellular entry of SARS-CoV-2. Curcumin also shows virucidal effects independently of SARS-CoV-2 variants and inhibitory effects on the release of pro-inflammatory cytokine IL-1b, IL-6, and IL-8 [106]. The curcumin of subtoxic concentrations is also effective in neutralizing SARS-CoV-2 in Calu-3 cells (human lung cancer cell line) and Vero E6 cells (African green monkey kidney cells) [107].

In a case series by Chabot and Huntwork [48], a 25-year-old male with COVID-19 diagnosed with the RT-PCR test took a capsule containing 1000 mg turmeric extract (95% curcuminoid) and 10 mg black pepper extract. Although the patient had suffered from ageusia persisting for 46 days, he experienced a complete recovery of taste 10 min after treatment. Another 28-year-old male COVID-19 patient complaining of ageusia took a capsule containing 1000 mg turmeric extract (95% curcuminoids), 15 mg black pepper extract, and 1000 mg *Boswellia serrata* plant extract. His taste sensation was improved 12 h after treatment and completely restored 3 days later. These outcomes suggest that the oral administration of curcumin with the extract of black pepper is effective for treatment of COVID-19-associated taste disorders. Piperin, a phytochemical component in black pepper, is considered to enhance the bioavailability of curcumin.

In addition to curcuminoid, phytochemicals belonging to flavonoid and capsaicinoid interact with the SARS-CoV-2 spike protein, host cell ACE2 receptor, RBD/ACE2 complex, and viral M^pro^, and exhibit antiviral, anti-inflammatory, neuroprotective, and antioxidant activity. In the literature, however, they have not been applied to the treatment of taste disorders of COVID-19 patients.

#### 3.1.6. Herbal Medicine

Multiple Pathogenic Mechanisms

Different classes of phytochemicals have antiviral [99], anti-inflammatory [100], neuroprotective [101], antioxidant [102], and SARS-CoV-2 M^pro^ inhibitory potential [103], which supports the rationale for herbs to be used for treating COVID-19-associated taste disorders.

2.Treatment with Herbal Medicine and Outcomes

Since the global outbreak of COVID-19, traditional herbal medicines have been actively used for its treatment in individual countries/regions. In the literature, however, their clinical applications to taste disorders of COVID-19 patients and survivors remain little known, except for Ayurveda. Ayurveda, which is an alternative medicine with historical roots in India, has a high potential to treat and prevent COVID-19 as the Ayurvedic regimens, using a combination of different medical herbs, kadha (herbal tea), golden milk (turmeric milk tea), nutritional supplements, and natural healing therapies, produce preventive and therapeutic effects on pre- and post-COVID-19 manifestations [108].

Ayurvedic treatments include a combination of natural healing methods and herbs. Wanjarkhedkar et al. [49] assessed the efficacy of an Ayurvedic regimen using *Dasamoolkaduthrayam Kashaya* and *Guluchyadi Kwatham*, ingredients of which possess antiviral and anti-inflammatory potentials [109]. Patients (n = 60, mean age: 44.0 years) who presented with ageusia due to mild to moderate COVID-19 diagnosed by a RT-PCR test orally received one tablet of 900 mg *Dasamoolkaduthrayam Kashaya* (consisting of extracts prepared from root, rhizome, or fruit of 14 different herbs) and one tablet of 600 mg *Guluchyadi Kwatham* (consisting of extracts prepared from stem, heartwood, stembark, or fruit of 5 different herbs) 12-hourly after a meal for 7 days in addition to the Standard of Care as per the Indian Council of Medical Research guidelines. The prevalence of ageusia was reduced from 75% on day 1 to 25% on day 3 and 3.3% on day 7, whereas the prevalence was 35.9% on day 3 and 25.6% on day 7 in control subjects (n = 39, mean age: 41.6 years) who received only the Standard of Care. Although bioactive flavonoids, terpenoids, and alkaloids have been suggested to be contained in medical herbs included in the Ayurvedic regimens recommended for COVID-19 treatments [108], phytochemical components in *Dasamoolkaduthrayam Kashaya* and *Guluchyadi Kwatham*, and their detailed compositions, have not been found in the literature.

#### 3.1.7. Nutraceutical

Association with Disease Severity and Multiple Pathogenic Mechanisms

There is evidence that low levels of nutraceuticals, especially vitamin D, are closely related to COVID-19 severity [110]. Vitamin D enhances ACE2 expression to restore the SARS-CoV-2 infection-disrupted balance between ACE/ACE2 and angiotensin II/angiotensin-(1–7), decreases pro-inflammatory cytokines, and increases anti-inflammatory cytokines [111]. In addition, vitamin D has a potent neuroprotective effect, linked to the regulation of neurotrophins that are responsible for the differentiation and maintenance of nerve cells [112]. Therefore, supplementation with vitamin D is expected to alleviate COVID-19-associated taste disorders.

2.Supplementation with Vitamin D and Outcomes

Sabico et al. [50] assessed the effects of vitamin D3 supplementation on COVID-19 symptoms with a multi-center randomized clinical trial. They orally administered 5000 IU and 1000 IU vitamin D3 to patients (n = 36, female: 41.7%, mean age: 46.3 years) and (n = 33, female: 60.6%, mean age: 53.5 years), respectively, once a day for 2 weeks. The patients had mild to moderate COVID-19 diagnosed by the RT-PCR test. The group of 5000 IU vitamin D3 showed a significantly shorter time to recover from ageusia (mean 11.4 days) compared with the group of 1000 IU vitamin D3 (mean 16.9 days). Decreases in IL-6 levels in the blood were also found in both groups.

Gaylis et al. [51] designed the formula for a nutraceutical supplement to treat persistent COVID-19 symptoms. In their clinical application, subjects (n = 51, female: approximately 67%, age: 21–73 year) who had suffered from different symptoms including ageusia for at least 3 months after SARS-CoV-2 infection orally received a supplement twice a day for 2 and 4 weeks. The nutraceutical formula consisted of vitamin D (1000 IU), β-caryophyllene (40 mg), pregnenolone (40 mg), dehydroepiandrosterone (30 mg), bromelain (416 mg, 2400 gelatin digesting unit/g), St. John’s Wort extract (150 mg), *Boswellia serrata* gum/resin extract (100 mg), quercetin (40 mg), and zinc as zinc picolinate (12 mg). Taste/smell loss in COVID-19 survivors became significantly milder 2 weeks after treatment, and the symptoms were further improved in 72–84% of subjects after 4 weeks.

#### 3.1.8. Photobiomodulation

Multiple Pathogenic Mechanisms

Photobiomodulation uses light-emitting diodes to produce red or near-infrared light to locally illuminate tissues subject to the treatment. Although photobiomodulation has been known about for a long time, it still has not gained general acceptance as an effective therapy because of uncertainty about its molecular and cellular mechanisms. In mechanistic considerations, the primary chromophore or photoacceptor is cytochrome c oxidase (unit IV in the mitochondrial respiratory chain), which absorbs light in the red or near-infrared region with resultant increases in the enzyme activity, mitochondrial membrane potential, and adenosine triphosphate production [113]. Photobiomodulation was also suggested to stimulate cell proliferation and differentiation, decrease pro-inflammatory cytokines such as IL-6, IL-1β, and TNF-α, increase neurogenesis and synaptogenesis, promote tissue repair, modulate the immune system, and inhibit apoptosis [114,115]. Therefore, photobiomodulation therapy without observed side-effects has recently been attracting attention as a possible approach to COVID-19 treatment [116]. Another class of photoacceptors in photobiomodulation is light-gated ion channels, such as the transient receptor potential (TRP) family of calcium channels [113]. Among TRP channels, transient receptor potential vanilloid 1 (TRPV1) is expressed in taste buds and influences bitter, sour, and salty taste qualities [117]. The applicability of photobiomodulation to COVID-19-associated taste disorders is supported by a case series that revealed the effectiveness of photobiomodulation in the management of oral complications related to cancer therapy, such as taste alteration (dysgeusia) [118].

2.Photobiomodulation and Outcomes

de Souza et al. [52] conducted a photobiomodulation therapy for a 34-year-old female who was infected with SARS-CoV-2 and developed ageusia. The patient was illuminated with three laser beams of 680 nm and three laser beams of 808 nm, which were applied for 2 min on the back of her tongue and the skin surface of her cheeks with the mouth slightly open to permit the light to reach the sides of her tongue and the inner mucosae of her cheeks. The treatment consisted of 10 sessions performed over 25 days, with a minimum interval of 48 h between sessions. The taste of the patient improved with each session and was back to normal after the last session.

In a pilot clinical study by Panhoca et al. [53], Caucasian patients (n = 20) with ageusia induced by COVID-19 were subjected to intraoral illumination with dual wavelengths (660 nm and 808 nm), 100 mW, three points, and 216 J/session. After 12 sessions, taste perception was significantly improved in patients’ self-reported evaluations.

#### 3.1.9. Alternative Medicine

Multiple Pathogenic Mechanisms

Acupuncture is an ancient practice of traditional Chinese medicine. It involves the insertion of extremely thin, solid, metallic needles into intradermal or subdermal loci by sticking the needles in specific points (acupuncture points). Since acupuncture exhibits an anti-inflammatory effect and restores the imbalanced autonomic nervous system, it may be an alternative medicine for COVID-19-associated taste disorders. Acupuncture is expected to be effective and safe treatment, especially for long-term persistent COVID-19 symptoms, as reviewed by Williams [119]. Acupuncture is used alone or in combination with another traditional medicine, moxibustion. Moxibustion is a heat therapy that usually bakes acupoints, specific sites on or very near the surface of the skin, by burning the dried plant mugwort, called moxa. Its therapeutic mechanism is related to thermal, radiation, and pharmacological (due to phytochemicals derived from moxa) effects [120].

2.Acupuncture and Moxibustion and Expected Outcomes

Chao et al. [121] reported a meta-analysis about the efficacy and safety of acupuncture in the treatment of post-COVID-19 taste disorders by collecting randomized controlled trials of acupuncture. They described acupuncture being effective for taste disorders, but there was still not sufficient evidence to support its treatment efficacy in sequelae of taste dysfunction. Luo et al. [122] conducted a systematic review and meta-analysis of traditional Chinese medicine combined with moxibustion for the treatment of COVID-19 sequelae, including taste disorders. They claimed that more large-scale high-quality trials should be carried out to confirm the efficacy and safety. While both studies suggested that acupuncture and moxibustion may be applicable to COVID-19 survivors or long-lasting COVID-19 symptoms (so-called “long COVID”), they mentioned no details about ageusia/dysgeusia or clinical outcomes.

### 3.2. Treatments of COVID-19-Associated Saliva Secretory Disorders

In addition to head and neck radiotherapy for cancer patients and autoimmune disorders such as Sjögren’s syndrome, certain medications are related to a reduction in salivary secretion. Antiviral remdesivir, hydroxychloroquine, lopinavir, and ritonavir have been widely prescribed for COVID-19 patients, and antihypertensive drugs are used by COVID-19 patients with hypertension. The administration of these drugs is the primary reason to cause xerostomia. Xerostomia or dry mouth is also known as a symptom in diabetes mellitus, which is one of the most common comorbidities of COVID-19 patients. Apart from these causal factors, SARS-CoV-2 infection is closely associated with saliva secretory disorders.

Saliva in the tongue film, a mucosal film covering the dorsum of the tongue, influences taste sensations by solubilizing, diluting, and modifying tastants. A significant number of patients infected with SARS-CoV-2 develop xerostomia, dry mouth, or hyposalivation, along with ageusia, dysgeusia, or hypogeusia. Saliva secretory disorders are secondary to taste disorders, and vice versa [123]. The combined taste and chewing function are associated with the production and secretion of saliva, and reduced salivary secretion influences taste perception. The pathogenesis and possible treatment may be at least partly common to both taste and saliva secretory disorders in COVID-19. Diverse therapeutic approaches have been tried to treat xerostomia, dry mouth, or hyposalivation induced by SARS-CoV-2 infection. In the following Sections, different types of treatments applicable to COVID-19-associated saliva secretory disorders are reviewed together with discussions on their pathogenic mechanisms and treatment rationales. They are summarized in Table 2, with the methods used, patients or subjects, expected clinical outcomes, and references included.

#### 3.2.1. Corticosteroid

Viral Cellular Entry and Inflammation of Salivary Glands

SARS-CoV-2 cellular entry-relevant ACE2, TMPRSS2, and Furin are expressed in the major salivary glands of humans [55]. ACE2 was found to be distributed in the minor salivary glands of COVID-19 patients [136]. Biopsy specimens from patients who died of COVID-19 indicated that ACE2 and TMPRSS2 are localized in the ductal epithelia and serous acinar cells of parotid and submandibular glands and minor salivary glands, and that SARS-CoV-2 is present in ductal lining cell cytoplasm, acinar cells, and ductal lumens of submandibular and parotid glands [137]. SARS-CoV-2 infection was confirmed in parotid and submandibular salivary glands and minor salivary glands of COVID-19 patients [138]. Given the expression of viral cellular entry-relevant bio-factors and the high viral load in saliva, major and minor salivary glands are targeted for the direct infection and invasion of SARS-CoV-2, and saliva secreted from them is rich in SARS-CoV-2. Salivary glands are the potential reservoir for SARS-CoV-2, and droplets of saliva contaminated with SARS-CoV-2 are potentially one of the causes of COVID-19 transmission [139]. After entering the relevant cells via the ACE2-, TMPRSS2-, and Furin-mediated pathways, cytopathic SARS-CoV-2 could cause damage to salivary glands. SARS-CoV-2 infection induces salivary gland inflammation, resulting in sialadenitis in the acute phase of COVID-19 and chronic sialadenitis via fibrosis repairment [140]. Parotitis and submandibular gland sialadenitis are associated with COVID-19 [141,142]. The turnover of salivary gland acinar cells ranges from 50 to 125 days, and they are replaced in 6 months. Once salivary glands are damaged, they need several months to recover their secretory functions, resulting in the long-term persistence of xerostomia, dry mouth, or hyposalivation. SARS-CoV-2 also remains for a certain period in tissues of patients who recovered from COVID-19, as prolonged SARS-CoV-2 RNA shedding is observed for months after symptomatic relief [143].

Given salivary gland inflammation through SARS-CoV-2 cellular entry and the consequent secretory dysfunction, anti-inflammatory corticosteroid is usable for treating COVID-19-associated saliva secretory disorders.

2.Treatment with Corticosteroid and Outcomes

Díaz Rodríguez et al. [124] reported a clinical case of a 78-year-old female who was infected with SARS-CoV-2 and had suffered from very intense sensations of mouth dryness, tongue and palate lesions, and angular cheilitis since hospitalization. They prescribed nystatin solution rinses 4 times a day for 15 days for the intraoral lesions and ointments containing triamcinolone acetonide, neomycin, and nystatin for angular cheilitis. After treatment, not only had the lesions disappeared, but the dry mouth was improved along with an increase in salivary secretion. Although concomitantly used nystatin has a pro-inflammatory property [144], triamcinolone is effective in alleviating COVID-19 symptoms.

#### 3.2.2. Zinc

Zinc Deficiency Induced by SARS-CoV-2 Infection

Salivary secretion from submandibular and parotid glands is reduced in correlation with decreasing zinc levels in salivary glands [125]. Zinc deficiency decreases the activity of salivary gland carbonic anhydrase, which plays an important role in the production and secretion of saliva and the regulation of salivary pH [145]. Carbonic anhydrases are zinc-metalloenzymes with a high binding affinity for zinc and activity depending on zinc. Among different isozymes, carbonic anhydrase VI is localized in the serous acinar and ductal cells of human parotid and submandibular glands, and is secreted into saliva from them [146]. In SARS-CoV-2 infection-induced hypozincemia, zinc is redistributed from the blood to the liver at the expense of zinc in peripheral tissues [85], which could disturb intracellular zinc homeostasis or cause intracellular zinc dyshomeostasis in salivary glands. Zinc deficiency should have a negative impact on salivary secretion. Experimentally induced zinc deficiency in rats significantly decreased saliva secreted from submandibular glands with morphological changes [147]. When carbonic anhydrase VI-deficient patients orally received zinc of 100 mg once a day for 4–6 months, their parotid saliva showed significant increases in zinc and carbonic anhydrase VI [90].

Therefore, zinc supplementation to compensate for deficient zinc could be one of the treatment options for COVID-19-associated saliva secretory disorders.

2.Supplementation with Zinc and Expected Outcomes

Tanaka [125] orally administered 300 mg zinc sulfate once a day for 6 months to non-COVID-19 patients (n = 93) with xerostomia and hypogeusia. Consequently, oral symptoms were relieved in 57.9–72.7% of patients.

Lane et al. [126] assessed the effect of zinc supplementation on salivary secretion. Non-COVID-19 subjects (n = 10, female: 50%, age: 17–37 years) orally ingested 15 mg zinc acetate with milk every morning. After 5 weeks, the flow rate of stimulated parotid saliva was increased, along with an increase in zinc levels in blood.

Arbabi-kalati et al. [127] performed a double-blind randomized clinical trial in which non-COVID-19 patients (n = 25, female: 48%, age: 18–70 years) undergoing chemotherapy took three capsules (220 mg zinc sulfate) daily until the end of chemotherapy. At 2–20-week follow-ups, the intensity of xerostomia was significantly lower compared with age- and gender-matched control subjects.

Kim et al. [128] measured the salivary flow rate of non-COVID-19 subjects (n = 29) with hyposalivation who rinsed the mouth with 0.25% ZnCl_2_ solution for 3 min after resting for 5 min. The mouth rinsing increased the secretion of both unstimulated and mastication-stimulated saliva.

Although zinc supplementation has not been clinically applied so far to COVID-19 patients, its successful outcomes in non-COVID-19 cases are considered to support the efficacy of treatment with zinc supplements for COVID-19-associated saliva secretory disorders, especially in patients with zinc deficiency induced by SARS-CoV-2 infection [148].

#### 3.2.3. Antiviral Drugs

Viral Invasion of Salivary Glands and Induced Inflammation

SARS-CoV-2 can directly invade major and minor salivary glands to cause damage to them because ACE2, TMPRSS2, and Furin, essential for viral cellular entry, are expressed in salivary glands, as described in Section 3.2.1. Acute and chronic inflammatory responses to SARS-CoV-2 infection occur in salivary glands [140], with the resultant impairment of their secretory functions.

When targeting cytopathic SARS-CoV-2 in salivary glands, the use of antiviral drugs is one of treatment options for COVID-19-associated saliva secretory disorders.

2.Treatment with Antiviral Drugs and Outcomes

In a case report by Zhang et al. [129], a 79-year-old female patient was hospitalized due to moderate COVID-19 that was caused by the Omicron variant BA.2.0 of SARS-CoV-2. While anecdotal reports have suggested that the Omicron variant causes oral symptoms with relatively low frequency compared with other SARS-CoV-2 variants such as the Delta variant, she developed xerostomia together with a cough. They orally administered Paxlovid, two 150 mg tablets of nirmatrelvir and one 100 mg tablet of ritonavir, to the patient 12-hourly for 5 days. Paxlovid (Pfizer’s blockbuster antiviral) is an authorized oral prescription medicine consisting of nirmatrelvir and ritonavir, both of which show antiviral effects by inhibiting SARS-CoV-2 M^pro^. On day 3 of treatment, xerostomia was relieved along with an improvement of the cough, lung exudation lesions, and inflammatory parameters. Although side-effects of Paxlovid were not observed in this study, a bad aftertaste is known for Paxlovid, so-called “Paxlovid mouth”. In addition, dysgeusia is more likely to be reported by patients receiving Paxlovid compared with patients receiving other protease inhibitors [149], and by pregnant or lactating COVID-19 patients who were prescribed with Paxlovid [150].

#### 3.2.4. Photobiomodulation

Multiple Pathogenic Mechanisms

Through interaction with cellular entry-relevant ACE2, TMPRSS2, and Furin, SARS-CoV-2 can directly invade salivary glands, as described in Section 3.2.1. Consequently, SARS-CoV-2 infection induces acute and chronic inflammation in salivary glands, parotitis, and submandibular gland sialadenitis, which can cause damage to salivary glands and affect their secretory functions.

Photobiomodulation stimulates cell proliferation and differentiation, inhibits inflammation, promotes tissue repair, enhances immunity, and prevents apoptosis [114,115]. Photobiomodulation also activates TRP channels as a photoacceptor [113]. While TRP channels are expressed in salivary glands [151], the activation of salivary gland TRPV1 results in increases in the salivary flow and secretion [152], suggesting that TRPV1 is responsible for dry mouth. The applicability of photobiomodulation to COVID-19-associated saliva secretory disorders is supported by the effectiveness of photobiomodulation in the management of oral complications related to cancer therapy, such as oral dryness [118].

Photobiomodulation acts on salivary glands to increase the number of ducts, the mitosis of epithelial cells, the protein synthesis in salivary glands, the blood circulation in salivary glands, and the flow rate of saliva [151]. These effects are crucial for its use to treat COVID-19-associated saliva secretory disorders. Although no clinical applications to COVID-19 patients have been reported so far, successful outcomes for xerostomia and hyposalivation in non-COVID-19 patients suggest the efficacy of photobiomodulation in COVID-19 cases.

2.Photobiomodulation and Expected Outcomes

Photobiomodulation is performed by illuminating the tissues with low-level lasers (usually ranging from 0.05 to 0.5 W at the source) of a red light-wavelength ranging from 630 to 685 nm and an infrared-wavelength ranging from 780 to 970 nm [153,154]. Palma et al. [130] evaluated the effect of photobiomodulation on the persistent xerostomia of head and neck cancer patients (n = 29, female: 27.6%, age: ≥37 years) receiving radiotherapy. These non-COVID-19 patients were subjected to photobiomodulation with a diode laser (wavelength of 808 nm, power density of 0.75 W/cm^2^, output power of 30 mW, illuminated area of 0.04 cm^2^, mean dose per point of 7.5 J/cm^2^, illumination time per point of 10 s, energy per point of 0.3 J, and energy per session of 6.6 J). Six extraoral points were illuminated on each parotid gland, three extraoral points on each submandibular gland, and two intraoral points on each sublingual gland. The patients underwent two laser sessions weekly for 3 months, for a total of twenty-four sessions. Consequently, the mean flow rates of both unstimulated and stimulated saliva were significantly increased.

Ferrandez-Pujante et al. [131] recruited non-COVID-19 patients (n = 60) who developed xerostomia due to drug use (n = 47) and Sjögren’s syndrome (n = 13). The patients (n = 30, female: 93.3%, mean age: 65.4 years) underwent photobiomodulation, and the resulting changes in salivary flow were compared with control subjects (n = 30, female: 100%, mean age: 68.4 years) who underwent simulated treatments. A diode laser (wavelength of 810 nm) was externally bilaterally illuminated to the parotid gland on a continuous basis at a dose of 6 J/cm^2^ for 2 min and 24 s (24 cm^2^ × 6 J/cm^2^ = 144 s, total 1 W), and to the submandibular gland on a continuous basis at a dose of 6 J/cm^2^ for 1 min and 12 s (12 cm^2^ × 6 J/cm^2^ = 72 s, total 1 W). One weekly session was carried out for a total of 6 weeks. As a result of the photobiomodulation treatment, the patients showed a significant improvement in xerostomia compared with the control group.

#### 3.2.5. Sialagogue

Promotion of Salivary Secretion

Given the parasympathetic innervation of parotid, submandibular, and sublingual glands, and minor salivary glands, the stimulation of parasympathetic nerves should improve xerostomia, dry mouth, and hypogeusia. Parasympathomimetic or muscarinic agonists pilocarpine and cevimeline stimulate salivary glands to increase saliva in patients with xerostomia [155]. However, these sialagogues are likely to cause side-effects such as nausea, vomiting, increased urinary frequency, and headache.

Salivary secretion is promoted in response to taste, especially a sour taste [155]. Although acidic compounds stimulate salivary secretion, their long-term use increases the risk of tooth erosion. Low-concentration malic acid combined with xylitol and fluoride was recently reported to significantly reduce dental erosion potential compared with conventional citric acid-based sialagogues [156].

2.Treatment with Malic Acid Sialagogue and Expected Outcomes

Gómez-Moreno et al. [132,157,158] suggested the possibility that a 1% malic acid topical sialagogue may be applicable for the treatment of COVID-19-associated saliva secretory disorders. They conducted a randomized double-blind study to evaluate the efficacy of malic acid sialagogue in patients with xerostomia induced by using antihypertensive drugs [132]. Non-COVID-19 patients (n = 25, female: 56%, mean age: 54.3 years) received a topical sialagogue spray (*Xeros Dentaid*^®^ *spray*) containing 1% malic acid, 10% xylitol, and 0.05% sodium fluoride. The spray was administered on demand with a maximum of 8 doses per day for 2 weeks. After treatment, the flow rates of both unstimulated and stimulated saliva were significantly increased compared with the placebo group (n = 20, female: 45%, mean age: 51.8 years). In their following studies, a topical sialagogue spray of 1% malic acid used for 2 weeks promoted the secretion of both unstimulated and stimulated saliva to improve antidepressant-induced dry mouth [157], and xerostomia in elderly people [158].

#### 3.2.6. Artificial Saliva

Substitution for Saliva

The sensation of mouth dryness is perceived due to insufficient mucosal wetness that is caused by a decrease in the salivary film coating on oral tissue surfaces [159]. As a symptomatic therapy, artificial saliva is usable as a saliva substitute for patients with xerostomia, dry mouth, or hyposalivation.

2.Use of Artificial Saliva and Outcomes

Eduardo et al. [160] prescribed artificial saliva for dry mouth in COVID-19 patients in an intensive care unit. Unfortunately, they mentioned neither the treatment procedure nor the clinical outcome in detail. However, their retrospective study provides a perspective on the use of artificial saliva for COVID-19-associated saliva secretory disorders.

#### 3.2.7. Chewing Gum

Mechanical Stimulation of Salivary Glands

The mechanical stimulation of major salivary glands is expected to increase the amount of secreted saliva. Chewing gum, usually sugar-free or artificially sweetened, has been used for managing xerostomia, dry mouth, or hyposalivation as described in a review of Kapourani et al. [24], who also referred to commercially available chewing gums used for xerostomia and their characteristics.

2.Use of Chewing Gum and Expected Outcomes

Xerostomia or dry mouth is induced in patients with chronic hemodialysis. Ozen et al. [133] conducted a prospective randomized controlled study in which non-COVID-19 patients receiving hemodialysis for at least 6 months were allocated to the chewing gum group and the control group. Patients (n = 22, female: 63.6%, mean age: 61.7 years) chewed gum (of various brands) for 10 min 6 times a day and when feeling mouth dryness or thirst, and they were followed up for 3 months. Xerostomia was time-dependently decreased and the flow rate of unstimulated saliva was increased compared with control subjects (n = 22, female: 36.4%, mean age: 61.4 years) who did not chew any gum.

#### 3.2.8. Alternative Medicine

Multiple Pathogenic Mechanisms

Acupuncture has been attracting attention as an alternative medicine for xerostomia or dry mouth because it suppresses inflammation, activates parasympathetic nerves, restores the autonomic nervous balance, and stimulates salivary glands via the cranial nerves. All of these effects are useful for the treatment of COVID-19-associated saliva secretory disorders, especially for long-term persistent symptoms [119]. Acupuncture has been used in dentistry for managing various conditions such as dental pain, temporomandibular disorder, trigeminal neuralgia, and dry mouth [161].

2.Acupuncture and Expected Outcomes

Blom and Lundeberg [134] conducted a retrospective study on patients (n = 70, female: 57.1%, age: 33–82 years) who had xerostomia due to primary and secondary Sjögren’s syndrome (n = 25, female: 92.0%, age: 33–72 years), irradiation (n = 38, female: 31.6%, age: 37–82 years), and other causes (n = 7, female: 71.4%, age: 38–73 years). These non-COVID-19 patients were subjected to acupuncture given in two series (12 treatments in each series), and then their oral conditions were examined after 6 months and followed up for 3 years. Flow rates of both unstimulated and stimulated saliva were increased, and such effects were maintained for 3 years by additional acupuncture. As supported by a case report of multidisciplinary treatment [162], acupuncture also facilitates recovery from post-COVID syndrome (long-term effects of COVID-19). In a clinical trial by Johnstone et al. [135], non-COVID-19 patients (n = 18) with pilocarpine-resistant xerostomia following radiotherapy of head and neck malignancies received acupuncture that was provided to three auricular points and one digital point bilaterally. After treatment, xerostomia was relieved in some patients. A systematic review and meta-analysis by Wu et al. [163] indicated that acupuncture combined with moxibustion produces better effects on radiotherapy-induced xerostomia.

While previous studies demonstrated a therapeutic effect in non-COVID-19 cases, evidence is still lacking for the clinical utility of acupuncture (in combination with moxibustion) in COVID-19-associated saliva secretory disorders.

## 4. Conclusions

Different types of treatments with multiple modes of action are applicable to taste and saliva secretory disorders of COVID-19 patients and survivors. Their efficacy was supported by successful outcomes in COVID-19 cases and also in non-COVID-19 cases. Promising treatments include the use of tetracycline, corticosteroid, zinc supplementation, stellate ganglion block, phytochemical curcumin, traditional herbal medicine, nutraceutical vitamin D, and/or photobiomodulation for COVID-19-associated taste disorders, and the use of corticosteroid, zinc supplementation, antiviral drug, malic acid sialagogue, chewing gum, acupuncture, and/or moxibustion for COVID-19-associated saliva secretory disorders. However, the high-quality evidence has been insufficient to support their routine use in clinical practice, requiring further well-designed studies with a greater variety of COVID-19 cases.

At present, fully validated treatments are still lacking for COVID-19-associated ageusia/dysgeusia/hypogeusia and xerostomia/dry mouth/hyposalivation in the early stage of SARS-CoV-2 infection and after recovery from the disease. Therefore, it is necessary to use the most suitable treatments according to symptoms and patient characteristics. Appropriately selected treatment and oral healthcare should be provided to COVID-19 patients and survivors suffering from taste and saliva secretory disorders. An understanding of currently available treatment options is required for dental professionals because they may not only treat patients who were infected with SARS-CoV-2 or who had recovered from COVID-19, but also first become aware of their abnormal taste and salivary secretion. By doing so, dentists and dental hygienists could play a crucial role in managing COVID-19 oral symptoms, and contribute to improving the oral health-related quality of life of the relevant dental patients.

## Figures and Tables

**Table 1 dentistry-11-00140-t001:** Treatments applicable to COVID-19-associated taste disorders.

Type	Mechanism	Method	Subject	Outcome	Reference
Tetracycline	Antiviral Anti-inflammatory Neuroprotective Anti-apoptotic	Oral administration of either doxycycline (100 mg/day or 100 mg twice a day) or minocycline (50 mg/day, 100 mg/day, or 100 mg twice a day)	COVID-19 patients (n = 38, female: 52.6%, age: 21–67 years) with mild disease quarantined at home	Ageusia disappeared in all patients within 7 days after treatment	Gironi et al. [38]
Corticosteroid	Anti-inflammatory	Local application of triamcinolone oral paste (0.1% triamcinolone acetonide)	COVID-19 patients (n = 60, female: 25.0%, mean age: 50.9 years)	Sweet, bitter, salty, and sour taste were improved in 83.3–91.7% of patients on day 5 of treatment	Singh et al. [39]
		Oral administration of corticosteroid (10 mg/day for the first week and reduced to 5 mg in the second week)	COVID-19 patients (n = 34, female: 55.9% and n = 33, female: 57.6%; grouped according to the different duration of taste disorders)	At weekly follow-ups up to 3 months, all patients recovered from ageusia at the end of treatment without side-effects	Gamil et al. [40]
Zinc	Compensation for deficient zinc	Supplementation with 220 mg zinc sulfate (corresponding to elemental zinc of 50 mg) twice a day	COVID-19 patients (n = 49) with different disease severity	When followed up until the pharyngeal swabs became negative, the duration of tase function recovery was shortened compared with control subjects (n = 56)	Abdel-maksoud et al. [41]
		Taking lozenges of zinc citrate (corresponding to elemental zinc of 23 mg), zinc citrate/zinc gluconate (23 mg), or zinc acetate (15 mg) every 2–4 h	COVID-19 outpatients (n = 4)	All patients showed symptomatic and objective improvements	Finzi [42]
		Supplementation with elemental zinc of 25 mg twice a day for 15 days	Ambulatory and hospitalized COVID-19 patients (n = 231, female: 47.6%, mean age: 54.6 years)	The symptom duration was shortened	Ben Abdallah et al. [43]
		Taking 6–12 lozenges of zinc gluconate/citrate (corresponding to elemental zinc of 23 mg) or zinc acetate (corresponding to elemental zinc of 15 mg) once a day	COVID-19 patients (n = 28, female: 60.7%, mean age: 40 years)	Symptoms including ageusia were improved 7 days after treatment and zinc gluconate was better tolerated than zinc acetate	Finzi and Harrington [44]
		Supplementation with a combination of zinc, magnesium, and calcium	COVID-19 pregnant patients (n = 74)	Ageusia/anosmia was reported by 41.9% of patients with zinc treatment, but by 57.2% of patients without zinc treatment	Citu et al. [45]
Stellate ganglion block	Treatment of dysautonomia	Right-sided stellate ganglion block with a local anesthetic and left-sided stellate ganglion block 2 days later	COVID-19 patient, a 42-year-old female who recovered from the disease but had continued to suffer from dysgeusia and anosmia	Immediately after treatment, dysgeusia and anosmia were improved, and taste/smell functions were normal at 2-week follow-up	Liu and Duricka [46]
		Right-sided stellate ganglion block, followed by left-sided stellate ganglion block on the next day	COVID-19 patient, a 44-year-old female with taste and smell loss who contracted the disease approximately 8 months ago	Within minutes after treatment, dysgeusia was drastically improved and taste function was normal at 60-day follow-up	Liu and Duricka [46]
		Right-sided stellate ganglion block with 4 mL of 0.25% bupivacaine and left-sided stellate ganglion block after 3 days	COVID-19 patient, a 48-year-old female who had recovered from the disease 4 months before but had altered taste to various types of foods	Taste disorders were improved a few days after treatment	Chauhan et al. [47]
Phytochemical: Curcumin	Antiviral Anti-inflammatory Neuroprotective Anti-apoptotic Antioxidant	Oral administration of capsule containing 1000 mg turmeric extract (95% curcuminoids) and 10 mg black pepper extract	COVID-19 patient, a 25-year-old male with ageusia persisting for 46 days	The patient experienced the complete recovery of taste function 10 min after treatment	Chabot and Huntwork [48]
		Oral administration of capsule containing 1000 mg turmeric extract (95% curcuminoids), 15 mg black pepper extract, and 1000 mg *Boswellia serrata* plant extract	COVID-19 patient, a 28-year-old male complaining of ageusia	Taste sensation was improved 12 h after treatment and completely restored 3 days later	Chabot and Huntwork [48]
Traditional herbal medicine: Ayurveda	Antiviral Anti-inflammatory	Oral administration of one tablet of 900 mg *Dasamoolkaduthrayam Kashaya* and one tablet of 600 mg *Guluchyadi Kwatham* 12-hourly after meal for 7 days in addition to the Standard of Care as per the Indian Council of Medical Research guidelines	COVID-19 patients (n = 60, mean age: 44.0 years) with ageusia due to mild to moderate disease	The ageusia prevalence of 75% on day 1 was reduced to 25% on day 3 and 3.3% on day 7, whereas it was 35.9% on day 3 and 25.6% on day 7 in the control group (n = 39, mean age: 41.6 years) who received only the Standard of Care	Wanjark-hedkar et al. [49]
Vitamin D	Nutraceutical supplementation	Oral administration of either 5000 IU vitamin D3 or 1000 IU vitamin D3 once a day for 2 weeks	COVID-19 patients with mild to moderate disease: 5000 IU vitamin D3 for patients (n = 36, female: 41.7%, mean age: 46.3 years) or 1000 IU vitamin D3 for patients (n = 33, female: 60.6%, mean age: 53.5 years)	After receiving 5000 IU vitamin D3, the time to recovery from ageusia was significantly reduced to mean 11.4 days compared with mean 16.9 days for 1000 IU vitamin D3	Sabico et al. [50]
		Oral administration twice a day of 1000 IU vitamin D, 40 mg β-caryophyllene, 40 mg pregnenolone, 30 mg dehydroepiandrosterone, 416 mg bromelain, 150 mg St. John’s Wort extract, 100 mg *Boswellia serrata* gum/resin extract, 40 mg quercetin, and 12 mg zinc picolinate	COVID-19 patients (n = 51, female: approximately 67%, age: 21–73 year) suffering from various symptoms including ageusia for at least 3 months after SARS-CoV-2 infection	Taste/smell loss became significantly milder after 2 weeks and the symptoms were further improved in 72–84% of subjects after 4 weeks	Gaylis et al. [51]
Photobio-modulation	Stimulation of cell proliferation and differentiation Anti-inflammatory Increase in neurogenesis Immune modulation Apoptosis inhibition Promotion of tissue repair Activation of taste bud TRPV1	Illumination of 3 laser beams (680 nm) and 3 laser beams (808 nm) for 2 min on the back of the tongue and the skin surface of the cheeks, consisting of 10 sessions performed over 25 days with a minimum interval of 48 h between sessions	COVID-19 patient, a 34-year-old female with ageusia	Taste function was improved with each session and back to normal after the last session	de Souza et al. [52]
		Intraoral illumination of dual wavelengths (660 nm and 808 nm) at 3 points, treatment consisting of 12 sessions	COVID-19 patients (n = 20) with ageusia	Taste perception was significantly improved	Panhoca et al. [53]

**Table 2 dentistry-11-00140-t002:** Treatments applicable to COVID-19-associated saliva secretory disorders.

Type	Mechanism	Method	Subject	Outcome	Reference
Corticosteroid	Anti-inflammatory	Nystatin solution rinses 4 times a day for 15 days for intraoral lesions and ointments containing triamcinolone acetonide, neomycin, and nystatin for angular cheilitis	COVID-19 patient, a 78-year-old female who had suffered from mouth dryness, tongue and palate lesions, and angular cheilitis since hospitalization	Dry mouth and salivary secretion were improved along with the disappearance of intraoral lesions	Díaz Rodríguez et al. [124]
Zinc	Compensation for deficient zinc	Oral administration of zinc sulfate (300 mg/day) for 6 months	Non-COVID-19 patients (n = 93) with oral symptoms	Xerostomia and hypogeusia were relieved in 57.9–72.7% of patients	Tanaka [125]
		Oral ingestion of 15 mg zinc acetate with milk every morning	Non-COVID-19 subjects (n = 10, female: 50%, age: 17–37 years)	After 5 weeks, the flow rate of stimulated parotid saliva was increased along with an increase in blood zinc levels	Lane et al. [126]
		Taking 3 capsules (220 mg zinc sulfate) daily until the end of chemotherapy	Non-COVID-19 patients (n = 25, female: 48%, age: 18–70 years) undergoing chemotherapy	At 2–20-week follow-ups, the intensity of xerostomia was lower compared with control subjects	Arbabi-kalati et al. [127]
		Mouth rinsing with 0.25% ZnCl_2_ solution for 3 min	Non-COVID-19 patients (n = 29) with hyposalivation	Both unstimulated and mastication-stimulated saliva were increased	Kim et al. [128]
Antiviral drug	Antiviral Inhibition of SARS-CoV-2 M^pro^	Oral administration of Paxlovid (two 150-mg tablets of nirmatrelvir and one 100-mg tablet of ritonavir) 12-hourly for 5 days	COVID-19 hospitalized patient, a 79-year-old female with moderate disease complaining of xerostomia due to infection with the Omicron variant BA.2.0 of SARS-CoV-2	Xerostomia was relieved on day 3 of treatment	Zhang et al. [129]
Photobio- modulation	Stimulation of cell proliferation and differentiation Anti-inflammatory Increase in ducts and epithelial cell mitoses Increase in salivary gland protein synthesis Increase in salivary gland blood circulation Increase in salivary flow rate Activation of salivary gland TRPV1	Illumination of laser (808 nm) to 6 extraoral points on each parotid gland, 3 extraoral points on each submandibular gland, and 2 intraoral points on each sublingual gland: illumination for 10 s per point with 2 laser sessions weekly for 3 months (a total of 24 sessions)	Non-COVID-19 patients (n = 29, female: 27.6%, age: ≥37 years) with persistent xerostomia after radiotherapy of head and neck cancer	Flow rates of both unstimulated and stimulated saliva were significantly increased	Palma et al. [130]
		External bilateral illumination of laser (810 nm) to the parotid gland on a continuous basis for 2.4 min and to the submandibular gland on a continuous basis for 1.2 min: one weekly session carried out for a total of 6 weeks	Non-COVID-19 patients (n = 30, female: 93.3%, mean age: 65.4 years) developing xerostomia due to drug use or Sjögren’s syndrome	Xerostomia was significantly improved compared with control xerostomic subjects (n = 30, female: 100%, mean age: 68.4 years) with simulated treatments	Ferrandez-Pujante et al. [131]
Sialagogue: Malic acid	Promotion of salivary secretion	Topical application of *Xeros Dentaid*^®^ *spray* (1% malic acid, 10% xylitol, and 0.05% sodium fluoride) on demand with a maximum of 8 doses per day for 2 weeks	Non-COVID-19 patients (n = 25, female: 56%, mean age: 54.3 years) with xerostomia induced by using antihypertensive drugs	Flow rates of both unstimulated and stimulated saliva were significantly increased compared with a placebo group (n = 20, female: 45%, mean age: 51.8 years)	Gómez-Moreno et al. [132]
Chewing gum	Mechanical stimulation of salivary glands	Chewing gum for 10 min 6 times a day and when feeling mouth dryness or thirsty	Non-COVID-19 patients (n = 22, female: 63.6%, mean age: 61.7 years) with chronic hemodialysis to cause xerostomia	At 3-month follow-up, xerostomia was alleviated and the flow rate of unstimulated saliva was increased compared with control subjects (n = 22, female: 36.4%, mean age: 61.4 years) who did not chew any gum	Ozen et al. [133]
Alternative medicine: Acupuncture	Anti-inflammatory Activation of parasympathetic nerves Restoration of autonomic nervous balance Stimulation of salivary glands via the cranial nerves	Acupuncture performed by giving 24 treatments in 2 series (12 treatments in each series)	Non-COVID-19 patients (n = 70, female: 57.1%, age: 33–82 years) suffering from xerostomia due to Sjögren’s syndrome (n = 25, female: 92.0%, age: 33–72 years), irradiation (n = 38, female: 31.6%, age: 37–82 years), and other causes (n = 7, female: 71.4%, age: 38–73 years)	Flow rates of both unstimulated and stimulated saliva were increased after 6 months, and the additional acupuncture maintained such effects for 3 years	Blom and Lunde-berg [134]
		Acupuncture applied to 3 auricular points and 1 digital point bilaterally	Non-COVID-19 patients (n = 18) with pilocarpine-resistant xerostomia due to radiotherapy	Xerostomia was relieved in some patients	Johnstone et al. [135]

## Data Availability

Not applicable.

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
