# Peer review of "Treatments of COVID-19-Associated Taste and Saliva Secretory Disorders"

_dentistry, 2023, doi:10.3390/dj11060140_

Round 1

Reviewer 1 Report

Comments on Tsuchiya et al:

The aim of this manuscript is to update information on treatments being applicable to oral symptoms, in the early stage of SARS-CoV-2 infection and after recovery from the disease. Moreover, this manuscript aims to discuss the pathogenic mechanisms, which underlying this infection, by performing a literature search in PubMed, LitCovid, ProQuest, and Google Scholar.

This manuscript shows rich content, providing a deep insight for some works: the study is within the journal’s scope, and I found it to be well-written, providing sufficient information. Even if the manuscript provides an organic overview, with a densely organized structure and based on well-synthetized evidence, there are some suggestions necessary to make the article complete and fully readable. For these reasons, the manuscript requires major changes.

Please find below an enumerated list of comments on my review of the manuscript:

INTRODUCTION:

LINE 25: As of 10 March 2023, over 676.6 million individuals were globally infected with severe acute respiratory syndrome coronavirus 2 (SARS-CoV-2) and over 6.8 million patients with coronavirus disease 2019 (COVID-19) had died in the worldaccording to the 28 Johns Hopkins University and Medicine Coronavirus Resource Center. Please, consider adding a comma in this sentence.

LINE 29: The causative agent for COVID-19, severe acute respiratory syndrome coronavirus-2 (SARS-CoV-2) is an enveloped positive single-stranded RNA virus, with the most prominent viral genome of 8.4–12 kDa in size. There is a 5’ terminal in this viral genome, the central part of this genome, rich in open reading frames, which encodes proteins essential for virus replication. Instead, the 3’ terminal includes five structural proteins, Spike protein (S), membrane protein (M), nucleocapsid protein (N), envelope protein (E), and hemagglutinin-esterase protein (HE)The Spike protein is responsible for the pathogenesis in the human species since its receptor-binding domain (RBD) links to human cell surface receptor protein Angiotensin-Converting Enzyme-2 (ACE-2), encoded by the ACE2 gene. ACE-2 has a ubiquitous distribution in the organs with the consequence that SARS-CoV-2 infection may affect the lungs primarily, leading to respiratory failureHowever, this infection simultaneously involves several tissue and organs, from kidneys to the heart, blood vessels, liver, pancreas, immune system, with significant consequences also on the oral health (see, for reference: Torge, D.; Bernardi, S.; Arcangeli, M.; Bianchi, S. Histopathological Features of SARS-CoV-2 in Extrapulmonary Organ Infection: A Systematic Review of Literature. Pathogens 202211, 867. https://doi.org/10.3390/pathogens11080867). This is the major concern of this manuscript: there is a lack of information, regarding the genetic and molecular features of SARS-CoV-2, which underlying the multiple tropism of this virus for differenttissue and organs.

LINE 31: During COVID-19 pandemic, different populations in the world were locked down, by government decision, in order to slow down and stop the spread of the virus. Specifically, the containment measures of COVID-19 included school and workplace closures, stay at home orders, travel and outdoor activities restrictions, with severe limitations on people movements (see, for reference: Giovannetti F, Lupi E, Di Giorgio D, Scarsella S, Oliva A, Di Fabio D, Prata P, Petricca G, Valentini V. Impact of COVID19 on Maxillofacial Fractures in the Province of L'Aquila, Abruzzo, Italy. Review of 296 Patients Treated With Statistical Comparison of the Two-Year Pre-COVID19 and COVID19. J Craniofac Surg. 2022 Jun 1;33(4):1182-1184. doi: 10.1097/SCS.0000000000008468. PMID: 36041111; PMCID: PMC9232240).

LINE 34: In addition to the typical symptom of fever, cough, dyspnea, myalgia, and cardiomyopathy in the early stage of SARS-CoV-2 infection, COVID-19 is characterized by oral symptoms consisting of ageusia (taste loss), dysgeusia (taste impairment), or hypogeusia (taste reduction) [3] and xerostomia, dry mouth (subjective complaint of oral dryness), or hyposalivation (objective reduction of salivary flow) [4]. Please, reformulate and rewrite this sentence in a more fluent way.

LINE 46: Furthermore, during COVID-19 pandemic, the management of oral disorders exposed healthcare professionals to a high risk of contagion, due to the high quantity of aerosolized virus in this area (see, for reference: Cassoni A, Pucci R, Mangini N, Fadda MT, Battisti A, Giovannetti F, Terenzi V, Della Monaca M, Priore P, Raponi I, Valentini V. Head and Neck Cancer Treatment during COVID-19 Pandemic: A Central Experience in Rome. Emergency Management, Infection Prevention and Control. Cancers (Basel). 2020 Dec 24;13(1):33. doi: 10.3390/cancers13010033. PMID: 33374237; PMCID: PMC7795055).

LINE 49: Many COVID-19 survivors report persistent taste disorders when followed up 1–12 months after symptom onset, disease diagnosis, or hospitalization [13,14]. The authors should reformulate this sentence in a more fluent way.

The main topic is interesting, and certainly of great clinical impact. As regards the originality and strengths of this manuscript, this is a significant contribute to the ongoing research on this topic, as it extends the research field on the treatments being applicable to such oral symptoms, in the early stage of SARS-CoV-2 infection and after recovery from the diseaseOverall, the contents are rich, and the authors also give their deep insight for some works.

As regards the section of methods, there is a specific and detailed explanation for the methods used in this study: this is particularly significant, since the manuscript relies on a multitude of methodological and statistical analysis, to derive its conclusions. The methodology applied is overall correct, the results are reliable and adequately discussed.

The conclusion of this manuscript is perfectly in line with the main purpose of the paper: the authors have designed and conducted the study properly. As regards the conclusions, they are well written and present an adequate balance between the description of previous findings and the results presented by the authors.

In conclusion, this manuscript is densely presented and well organized, based on well-synthetized evidence. The authors were lucid in their style of writing, making it easy to read and understand the message, portrayed in the manuscript. Besides, the methodology design was appropriately implemented within the study. However, many of the topics are very concisely covered. This manuscript provided a comprehensive analysis of current knowledge in this field. Moreover, this research has futuristic importance and could be potential for future research. However, major concerns of this manuscript are with the introductive section: for these reasons, I have major comments for this section, for improvement before acceptance for publication. The article is accurate and provides relevant information on the topic and I have some major points to make, that may help to improve the quality of the current manuscript and maximize its scientific impact. I would accept this manuscript if the comments are addressed properly.

Reviewer 2 Report

Overall, this is a well written paper on the taste and saliva secretory disorder due to COVID-19. Although there have been many papers published focusing on the olfactory and taste dysfunction due to COVID-19, a review on saliva secretory disorder is rather few. This will be an important information for people in the field.

The tables are split at a strange place leaving blank space between and continuing on the next page. I know this happens and frustrating, but it is better to fix this as it does not look good.

The part that I am most concerned is the description on “Traditional herbal medicine: Ayurveda”. I am not against traditional herbal medicine but why did you select only Ayurveda here? There are traditional herbal medicines in many countries such as in China and Japan, and it will be fair to choose at least a couple of more other countries. Also, it will be more convincing if you write in detail at the chemical compounds level for these herbal medicines to ensure they are not pseudoscience. 

Line 451-459: Photobiomodulation is a rarer type of treatment. It will help readers if you add mechanistic insights of this treatment. The same to other types of treatments as well, i.e., wherever possible, please add description on how they are working, or considered to be working. Examples of cases are not enough.

Listing cases that the treatment methods worked will not be convincing enough for academic journals. Companies selling medicines/supplements can do so and it becomes that level. The differences in the treatments are interesting to summarize and it will be important information, however it is nice to add more information on what are, or what could be, the mechanisms.

Minor comments:

Line 399: “TR-PCR” -> “RT-PCR”

Reviewer 3 Report

The scientific paper "Treatments of COVID-19-Associated Taste and Saliva Secretory Disorders" aimed to update information on treatments being applicable to such oral symptoms in the early stage of SARS-CoV-2 infection and after recovery from the disease and to discuss the pathogenic mechanisms underlying them.

It can be considered that:

1)      The author could increase the abstract by including the main findings and the conclusion, more clearly;

2)      In the introduction, more details could be inserted about the anatomical and physiological relationships of the loss of smell and taste. I suggest reading and, if you want, the inclusion of doi: 10.5501/wjv.v11.i5.362.

3)      In the photobiomodulation section (4.4), more details about recent research in the area could be inserted. I suggest reading and, if you wish, including doi: 10.3390/life11060580.

4)      In the general context, the review was very well conducted by the author, the tables very complete in their content and, therefore, of great relevance for readers.

Minor editing of English language required

Round 2

Reviewer 1 Report

The reviewer has been a little disappointed in seeing the most part of the suggestions not complied. The phenomena of the SARS-COV-2 is relatively new and including all the information regarding would be helpful for all the readers. However the other suggestions were complied. If the author would like to reconsider the position on its replies is welcomed. 

Reviewer 2 Report

This is a well-written paper with many information that will be useful for researchers and medical doctors who treat patients with COVID-19-induced taste dysfunction and salivary gland dysfunction. I have just few more additional comments.

Line 149-151: I believe the role of TLR and TRPV1 as receptors for SARS-CoV-2 as entry site is not established yet and it is at the possibility level from the binding affinity, thus it is better to write in that way. TLRs recognize the virus, which triggers production of proinflammatory cytokines and chemokines, involved in the problematic immune responses when it is excessive or prolonged (https://www.futuremedicine.com/doi/10.2217/fvl-2021-0249) . The binding, especially with TLR4, is considered to increase the expression of ACE2, which will increase the entry rate. I have not found papers reporting that TLR4 becomes the entry site yet but I may have missed some papers. Here

About binding affinity with TLR4: https://www.frontiersin.org/articles/10.3389/fmicb.2022.948770/full

About the influences of binding to TLR4 (please note that they say ‘ “may” bind and activate TLR4’ ): 

https://www.ncbi.nlm.nih.gov/pmc/articles/PMC7811571/#:~:text=Evidence%20for%20Direct%20Binding%20of,severe%20stages%20of%20COVID%2D19.

About the influence of TLR activation in response to SARS-CoV-2:

https://www.ncbi.nlm.nih.gov/pmc/articles/PMC8622567/

About the TRPV1, it is a different story. There are studies suggesting that TRPV1 "may" serve as entry sites not only to SARS-CoV-2 but also to some other respiratory viruses. For example, https://www.ncbi.nlm.nih.gov/pmc/articles/PMC8599155/. So, although I would still say it is still at the suggested level, there are possibilities that it serves as entry sites. This is actually a very interesting point that is not discussed as much as the ACE2 and other proteins serving as entry sites such as NPR1 and sialic acid, as just a couple of examples.

What this also suggests is that, if they serve as receptors and entry sites and considering that there are many phytochemicals with binding affinity with TRP channels (https://www.ncbi.nlm.nih.gov/pmc/articles/PMC4240255/), this may explain one of the mechanisms that phytochemicals facilitate recovery from COVID-19, i.e., binding by the phytochemicals might be contributing to suppressing the further entry by the virus. ( https://www.ncbi.nlm.nih.gov/pmc/articles/PMC8217345/ )

Very interesting.

Line 530-533: This description on anecdotal report is non-scientific. Please remove. Mask helps to keep humidity. https://www.cell.com/action/showPdf?pii=S0006-3495%2821%2900116-8
